# Chest Compression Rates of 90/min versus 180/min during Neonatal Cardiopulmonary Resuscitation: A Randomized Controlled Animal Trial

**DOI:** 10.3390/children9121838

**Published:** 2022-11-28

**Authors:** Marlies Bruckner, Mattias Neset, Catalina Garcia-Hidalgo, Tze-Fun Lee, Megan O’Reilly, Po-Yin Cheung, Georg M. Schmölzer

**Affiliations:** 1Centre for the Studies of Asphyxia and Resuscitation, Neonatal Research Unit, Royal Alexandra Hospital, Edmonton, AB T5H 3V9, Canada; 2Department of Pediatrics, Faculty of Medicine and Dentistry, University of Alberta, Edmonton, AB T6G 2R3, Canada; 3Division of Neonatology, Department of Pediatrics and Adolescent Medicine, Medical University of Graz, 8036 Graz, Austria

**Keywords:** infant, newborn, chest compression, resuscitation

## Abstract

Background: To compare chest compression (CC) rates of 90/min with 180/min and their effect on the time to return of spontaneous circulation (ROSC), survival, hemodynamic, and respiratory parameters. We hypothesized that asphyxiated newborn piglets that received CC at 180/min vs. 90/min during cardiopulmonary resuscitation would have a shorter time to ROSC. Methods: Newborn piglets (n = 7/group) were anesthetized, intubated, instrumented and exposed to 45 min normocapnic hypoxia followed by asphyxia and cardiac arrest. Piglets were randomly allocated to a CC rate of 180/min or 90/min. CC was performed using an automated chest compression machine using CC superimposed with sustained inflation. Hemodynamic and respiratory parameters and applied compression force were continuously measured. Results: The mean (SD) time to ROSC was 91 (34) and 256 (97) s for CC rates of 180/min and 90/min, respectively (*p* = 0.08). The number of piglets that achieved ROSC was 7 (100%) and 5 (71%) with 180/min and 90/min CC rates, respectively (*p* = 0.46). Hemodynamic parameters (i.e., diastolic and mean blood pressure, carotid blood flow, stroke volume, end-diastolic volume, left ventricular contractile function) and respiratory parameters (i.e., minute ventilation, peak inflation and peak expiration flow) were all improved with a CC rate of 180/min. Conclusion: Time to ROSC and hemodynamic and respiratory parameters were not statistical significant different between CC rates of 90/min and 180/min. Higher CC rates during neonatal resuscitation warrant further investigation.

## 1. Introduction

Current neonatal resuscitation guidelines recommend a 3:1 compression to ventilation (C:V) ratio with 90 chest compressions (CCs) and 30 inflations to achieve approximately 120 events per minute [1,2]. However, the optimal CC rate to optimize coronary and cerebral perfusion while providing adequate ventilation of an asphyxiated newborn remains unknown [3].

A decade ago, we described a different approach to chest compression, by superimposing a sustained inflation (SI) during chest compression (CC + SI) [4]. The observed benefits included passive lung aeration resulting in improved minute ventilation and oxygenation and improved hemodynamics with significantly higher pulmonary and carotid blood flow [4]. Schmölzer et al. compared CC + SI with 3:1 C:V and reported significant reduced time to return of spontaneous circulation (ROSC) [38 (23–44) s vs. 143 (84–303) s (*p* = 0.0008)] and improved survival [7/8 [87.5%] vs. 3/8 [37.5%] (*p* = 0.038)] in bradycardic piglets [5]. We further compared CC + SI with chest compression rate of 90/min or 120/min and observed similar time of ROSC, survival rates, and hemodynamic and respiratory parameters during cardiopulmonary resuscitation [6]. However, when we compared CC + SI with a CC rate of 90/min with 3:1 C:V, CC + SI significantly reduced the median (IQR) time of ROSC [34 (28–156) s vs. 210 (72–300) s, (*p* = 0.048)] [7]. While these studies show, that CC + SI at either rates of 90/min or 120/min reduced time to ROSC and improves survival, a mathematical study suggests that the most effective CC rate to optimize systemic perfusion depends upon body size and weight [3]. This would translate to CC rates of 180/min in term infants and higher rates for preterm infants. However, CC rates of 180 CC/min or above are impossible to achieve using a 3:1 C:V ratio. If a clear effect of body size and weight on the optimal compression frequencies for CPR exists, then optimizing compression frequencies for neonatal CPR has the potential to improve short- and long-term outcomes in newborn infants.

We aimed to examine different CC rates and their effect on the time to return of spontaneous circulation (ROSC), survival, and hemodynamic and respiratory parameters. We hypothesized that a CC rate of 180/min compared to 90/min during cardiopulmonary resuscitation would result in a shorter time to ROSC in asphyxiated newborn piglets.

## 2. Materials and Methods

All experiments were conducted between January and November 2020 in accordance with the guidelines and approval of the Animal Care and Use Committee (Health Sciences), University of Alberta (AUP00001764), presented according to the ARRIVE guidelines [8], and registered at preclincialtrials.eu (PTCE0000148). A graphical display of the study protocol is presented in Figure 1. The authors declare that all supporting data are available within the article.

### 2.1. Randomization

Piglets were randomly allocated to two groups (“CC rate 180/min or “90/min”). Randomization was 1:1 with variable block sizes using a computer-generated randomization program (http://www.randomizer.org, accessed on 24 November 2022). Sequentially numbered, sealed, brown envelopes containing the group allocation were opened during the experiment (Figure 1).

### 2.2. Sample Size and Power Estimates

The primary outcome measure was the time of CPR to achieve ROSC. Our previous studies showed a mean (standard deviation) ROSC of 220 (25) s with a CC rate of 90/min. We hypothesized that a CC rate of 180/min would reduce the time to achieve ROSC. A sample size of 7/group would be sufficient to detect a clinically important (20%) reduction in time to ROSC (i.e., 176 s vs. 220 s) with 90% power and a 2-tailed alpha error of 0.05.

### 2.3. Inclusion and Exclusion Criteria

Newborn mixed-breed piglets (0–3 days of age) obtained on the day of experimentation from the University Swine Research Technology Center were included. There were no exclusion criteria.

### 2.4. Animal Preparation

Piglets were instrumented as previously described with some modifications [5]. Following the induction of anesthesia using isoflurane, piglets were intubated via tracheostomy, and mechanical ventilation (Sechrist infant ventilator model IV-100; Sechrist Industries, Anaheim, CA, USA) was commenced at a 20/min rate, peak inflation pressure of 25 cmH_2_O and positive end-expiratory pressure of 5 cmH_2_O. Oxygen saturation was kept within 90–100%, glucose level and hydration were maintained with an intravenous infusion of 5% dextrose at 10 mL/kg/h. During the experiment, anesthesia was maintained with intravenous propofol 5–10 mg/kg/h and morphine 0.1 mg/kg/h. Additional doses of propofol (1–2 mg/kg) and morphine (0.05–0.1 mg/kg) were also given as needed, and their body temperature was maintained at 38.5–39.5 °C by using an overhead warmer and a heating pad.

### 2.5. Hemodynamic Parameters

A 5-French Argyle^®^ (Klein-Baker Medical Inc., San Antonio, TX, USA) double-lumen catheter was inserted into the femoral vein for fluid administration and medications. A 5-French Argyle^®^ single-lumen catheter was inserted above the right renal artery via the femoral artery for continuous arterial blood pressure monitoring and arterial blood gas measurements. The right common carotid artery was exposed and encircled with a real-time ultrasonic flow probe (2 mm; Transonic Systems Inc., Ithica, NY, USA) to measure carotid blood flow. A Millar catheter (MPVS Ultra, ADInstruments, Houston, TX, USA) was inserted into the left ventricle (LV) via the left common carotid artery for continuous measurement of stroke volume, end-diastolic volumes, dp/dt_max_ (maximal rate of rise of left ventricular pressure), and dp/dt_min_ (minimum rate of change of ventricular pressure), which served as a surrogate for cardiac output. Because of the size difference between the Millar catheter and LV longitudinal axis, which poses a limitation for the accuracy of in vivo volume measurement, an alpha factor = 0.46, based on comparison between Millar’s recording and direct echocardiographic measurements in three piglets, was used to correct the conductance volume [9].

Piglets were placed in the supine position and allowed to recover from surgical instrumentation until baseline hemodynamic measures were stable (minimum of one hour). The ventilator rate was adjusted to keep the partial arterial CO_2_ between 35–45 mmHg as determined by periodic arterial blood gas analysis. Arterial blood pressure, heart rate, and percutaneous oxygen saturation were continuously measured and recorded throughout the experiment with a Hewlett Packard 78833B monitor (Hewlett Packard Co., Palo Alto, CA, USA).

### 2.6. Respiratory Parameters

A respiratory function monitor (NM3, Respironics, Philips, Andover, MA, USA) continuously measured tidal volume, airway pressures, gas flow, and end-tidal CO_2_. The sensor was placed between the endotracheal tube and the ventilation device. Tidal volume was calculated by integrating the flow signal, and end-tidal CO_2_ was measured using a nondispersive infrared absorption technique [10,11]. The accuracy for gas flow was ±0.125 L/min, and the end-tidal CO_2_ was ±2 mmHg.

### 2.7. Automated Chest Compression Machine

The automated CC machine was custom designed in our laboratory. The settings for the automated CC machine were anterior-poster depth 33%, acceleration of compression 500 cm/s^2^, speed of recoil 50 cm/s, a simulated two-thumb technique, and a CC rate of 90/min or 180/min according to group allocation [12,13].

### 2.8. Force Measurement

A FlexiForce A201 sensor (TekScan, Boston, MA, USA) was placed on the bottom of the plunger of the automated CC machine to measure the applied compression force. The applied compression force was recorded with Arduino Software (Somervile, MA, USA) with a sample rate of 200 Hz [12,13].

### 2.9. Experimental Protocol

Piglets were randomized into two groups: “CC rate 180/min” or “CC rate 90/min”. Following surgical instrumentation and stabilization, the piglets were placed onto the automated CC machine, which was placed on the surgical bed. The piglets’ chest diameter was measured from the sternum to the vertebrae touching the bed (anterior to posterior) with a measuring tape, and then the anterior-posterior depth of 33% was calculated [12,13]. Piglets were then exposed to 45 min of normocapnic hypoxia, which was followed by asphyxia. Normocapnic hypoxia was achieved by reducing the fraction on inspired oxygen and adjusting the ventilation rate. Asphyxia was achieved by disconnecting the ventilator and clamping the endotracheal tube until asystole. Asystole was defined as zero carotid blood flow and no audible heartbeat during auscultation. Fifteen seconds after asystole, positive pressure ventilation was provided for 30 s with a Neopuff T-Piece (Fisher & Paykel, Auckland, New Zealand) with 21% oxygen, peak inspiratory pressure of 30 cmH_2_O, positive end-expiratory pressure of 5 cmH_2_O, gas flow of 8 L/min, and at a rate of 50/min. After 30 s of positive pressure ventilation, mechanical CC using our automated CC machine was initiated [12,13] using 21% oxygen [14,15], with an antero-posterior chest diameter depth of 33% [12,13], and continuous CC was initiated during sustained inflation (CC + SI) with a peak inspiratory pressure of 30 cmH_2_O for 30 s [16,17,18]. Sustained inflation was interrupted for 1 s before a further 30 s of sustained inflation was provided, and this was continued until ROSC [5]. Chest compressions were provided for a maximum time of 10 min, if no ROSC, resuscitation efforts were stopped. Epinephrine (0.02 mg/kg per dose) was administered intravenously 2 min after the start of positive pressure ventilation and every 3 min until ROSC with a maximum of three doses [1,2], as the maximum resuscitation time was 10 min. The administration of epinephrine was immediately followed by a saline flush of 3 mL [19,20]. ROSC was defined as an unassisted heart rate >100/min for at least 15 s. After ROSC, piglets recovered for one hour before being euthanized with an intravenous overdose of sodium pentobarbital (120 mg/kg). If there was no ROSC, piglets were euthanized immediately with an intravenous overdose of sodium pentobarbital (120 mg/kg).

### 2.10. Data Collection and Statistical Analysis

Demographics of study piglets were recorded. Transonic flow probe, heart rate and pressure transducer outputs were digitized and recorded with LabChart^®^ programming software (AD Instruments, Houston, TX, USA). To analyze the hemodynamic data until time to ROSC (i.e., arterial blood pressure, central venous pressure, and carotid blood flow), the duration of CC time was divided into 10 epochs. To analyze stroke volume, end-diastolic volume, dp/dt_max_, and dp/dt_min_, the pressure–volume loops were compared between groups. Airway pressures, gas flow, tidal volume, and end-tidal CO_2_ were measured and analyzed using Flow Tool Physiologic Waveform Viewer (Philips Healthcare, Wallingford, CT, USA). For all respiratory parameters, the median values for each piglet during CPR were calculated first, and then the mean of the median was calculated for comparison. Time to ROSC was the primary outcome measure, however when a piglet did not achieve ROSC (maximum CPR time of 10 min or 600 s), this time was used as time to ROSC in the analysis.

The data are presented as the mean (standard deviation—SD) for normally distributed continuous variables and median (interquartile range—IQR) when the distribution was skewed. The data were tested for normality (Shapiro–Wilk and Kolmogorov–Smirnov test) and compared using t-tests or rank sum for normally or skewed distributed data. *p*-values are 2-sided, and *p* < 0.05 was considered statistically significant. Statistical analyses were performed with SigmaPlot (Systat Software Inc., San Jose, CA, USA).

## 3. Results

Fourteen newborn mixed breed piglets (0–3 days, weight 1.9–2.4 kg) were randomly assigned to CC rate of 180/min (n = 7) or 90/min (n = 7). Parameters at baseline or at start of CPR (end of asphyxia) were not different between groups (Table 1). The median (IQR) duration of asphyxia was 470 (360–585) and 440 (280–506) s in the 180/min and 90/min CC groups, respectively (*p* = 0.65).

### 3.1. Resuscitation and Primary Outcome

The number of piglets that achieved ROSC was 7 (100%) and 5 (71%) with 180/min and 90/min CC rates, respectively (*p* = 0.46). When all piglets were included in the median (IQR) time to ROSC was 103 (79–170) s for CC rate of 180/min and 189 (96–600) s for CC rate of 90/min (*p* = 0.12). However, when the piglets who did not achieve ROSC were excluded the median (IQR) time to ROSC was 103 (79–170) s for CC rate of 180/min and 136 (88–395) s for CC rate of 90/min (*p* = 0.43). Piglets received 0 (0–1) and 1 (0–3) epinephrine boluses with CC rates of 180/min and 90/min, respectively. The mean (SD) compression forces were 2.4 (1) and 2.6 (0.7) kg at 90 CC/min and 180 CC/min, respectively.

### 3.2. Hemodynamic Parameters

Hemodynamic parameters at baseline and at commencement of resuscitation were not different (Table 1). Diastolic and mean blood pressure and carotid blood flow were significantly higher with a CC rate of 180/min, while systolic blood pressure was not different between groups (Figure 2). Stroke volume and end-diastolic volume were improved, but this did not reach statistical significance (Figure 3), while dp/dt_max_ and dp/dt_min_ significantly improved with a CC rate of 180/min (Figure 3).

### 3.3. Respiratory Parameters

The tidal volume was not different between groups, while the minute ventilation was significantly increased with CC rate 180/min with mean (SD) 945 (249) compared to 522 (79) mL/kg with CC rate 90/min group (*p* = 0.003) (Table 2). Similarly, peak inflation flow and peak expiration flow were significantly higher with a CC rate of 180/min (Table 3).

Data separated by sex are presented in Table 3.

## 4. Discussion

Current neonatal resuscitation guidelines recommend providing 90 CC and 30 inflations (=120 events/min) to optimize cardiac output and oxygen delivery [1,2]. However, the optimal CC rate during neonatal CPR remains unclear. A mathematical study suggests that the most effective CC rate depends upon body size and weight and that higher CC rates, as currently recommended, might improve survival in newborn infants [3]. In the current study, we compared CC rates of 180/min and 90/min using CC + SI. The results can be summarized as follows: During CC + SI with a CC rate of 180/min (i) survival and (ii) time to ROSC was not different to CC + SI with a CC rate of 90/min; (iii) less epinephrine administration; (iv) higher blood pressure (diastolic and mean) and carotid blood flow (Figure 2); (v) similar stroke volume and end-diastolic volume, and significantly improved left ventricular function (Figure 3); and (vi) significantly increased minute ventilation and thereby oxygen delivery (Table 2).

Mathematical models described the maximal cardiac output during CPR depends on the fraction of cycle time available for pump filling and emptying, with pump filling being dominant [3,21]. Fitzgerald et al. compared CC rates between 20–140/min in 6–16 kg adult mongrel dogs and reported the maximum cardiac output would be achieved with a CC rate of 126/min [21]. The physiological heart rate in neonates ranges between 120–160/min [22], and an increase in the CC rate might have the potential to boost artificial cardiac output compared to recommended CC rates, which are based largely on experimental work in animal models larger than neonates. The mathematical model by Babbs et al. calculated the optimal CC rate for maximum cardiac output with a CC rate of 184/min for a 3 kg newborn infant, indicating an effect of body size and weight [3]. In the current study, stroke volume was higher at a CC rate of 180/min (Figure 3), and as cardiac output is simple, the product of compression rate × stroke volume, cardiac output was also higher (results not shown).

During neonatal resuscitation, Patel et al. randomized asphyxiated transitioned newborn piglets to continuous CC with asynchronized ventilation with CC rates of 90/min, 100/min, and 120/min and reported similar mean times to ROSC [23]. Notably, the piglets in the 90/min and 100/min groups had higher cerebral inflammation and brain injury than those in the 120/min group [23]. Similar, Vali et al. [24] used a transitional asphyxia lamb model and randomized them to either 3:1 C:V (90 CCs + 30 breaths/min) or continuous CC with asynchronized ventilation with CC rate of (120 CCs + 30 breaths/min) and reported no difference in rate of and time to ROSC. Continuous CC with asynchronized ventilation was associated with a higher PaO_2_, greater left carotid blood flow, and oxygen delivery. We recently compared CC + SI with 3:1 C:V in bradycardic piglets and reported significant reduced time to return of spontaneous circulation (ROSC) [38 (23–44) s vs. 143 (84–303) s (*p* = 0.0008)] and improved survival [7/8 [87.5%] vs. 3/8 [37.5%] (*p* = 0.038)] [5]. Similar, using CC + SI with either a CC rate of 90/min or 120/min resulted in similar time of ROSC, survival rates, and hemodynamic and respiratory parameters. In comparison, when CC + SI with a CC rate of 90/min with 3:1 C:V was compared a significantly reduced the median (IQR) time of ROSC with CC + SI was observed [34 (28–156) s vs. 210 (72–300) s, (*p* = 0.048)] [6]. However, CC with a higher rate had higher cardiac output and left ventricular function expressed by dp/dt_max_ and dp/dt_min_ [6]. Similarly, we recently compared cardiac function (i.e., stroke volume, cardiac output, and left ventricular function (i.e., dp/dt_max_ and dp/dt_min_)) with CC rates between 60–180/min [25]. Stroke volume and cardiac output were highest with a CC rate of 180/min, while end-diastolic volume, dp/dt_max_ or dp/dt_min_ was highest at a CC rate of 150/min. A further increase to a CC rate of 180/min did not further increase the end-diastolic volume, dp/dt_max_ or dp/dt_min_ [25]. These data suggest that CC with a rate of 150–180/min might have optimal cardiovascular performance.

We used a custom-designed chest compression machine [12,13], which allowed consistent delivery of CC rates and reduced potential bias (e.g., fatigue during CC or inability to constantly achieve allocated rate). Indeed, manikin studies compared CC rates of 90/min vs. 120/min and observed higher fatigue and up to 50% decay in CC depth after 90 or 120 s CC rate, respectively [26,27]. While we recognize that CPR using a CC rate of 180/min using a 3:1 C:V ratio is nearly impossible, it might be feasible using continuous compressions.

### Limitations

In the current study, we used continuous CC during sustained inflation [4], which is mentioned in the knowledge gap section of the neonatal resuscitation guidelines, however it is currently not recommended [1,2]. Our use of a piglet asphyxia model is a great strength of this translational study, as this model closely simulates delivery room events, with the gradual onset of severe asphyxia leading to bradycardia. Our asphyxia model uses piglets that have already undergone the fetal-to-neonatal transition, and piglets were sedated/anesthetized. Our model uses tracheostomy with a tightly sealed endotracheal tube to prevent endotracheal tube leak; this may not occur in the delivery room, as uncuffed endotracheal tubes are routinely used. A strength of this study is the use of our automated chest compression machine, which can apply high rates of CC [12,13]^.^

## 5. Conclusions

Time to ROSC and survival were not statistically different between CC rates of 90/min and 180/min. Respiratory and hemodynamic parameters were also improved with a CC rate of 180/min compared to 90/min. A higher CC rate might improve organ perfusion and oxygen delivery compared to lower CC rates and warrants further investigation.

## Figures and Tables

**Figure 1 children-09-01838-f001:**
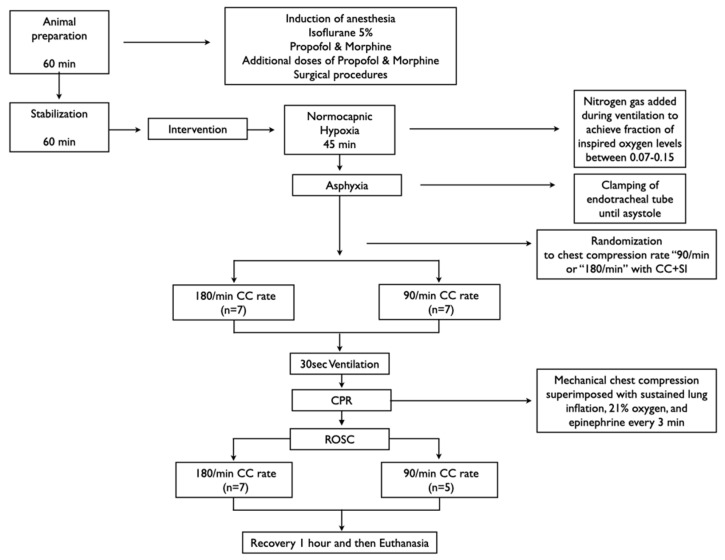
Study flow diagram.

**Figure 2 children-09-01838-f002:**
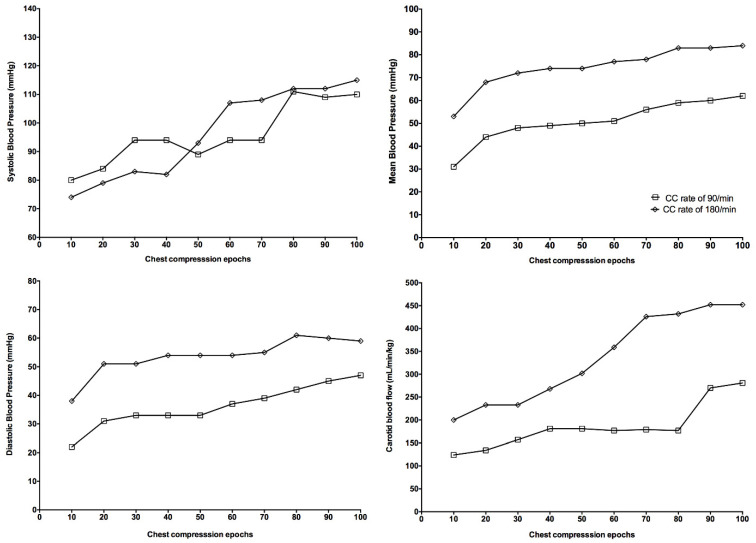
Changes in systolic blood pressure, diastolic blood pressure, mean blood pressure, and carotid blood flow during chest compression were divided into 10 epochs.

**Figure 3 children-09-01838-f003:**
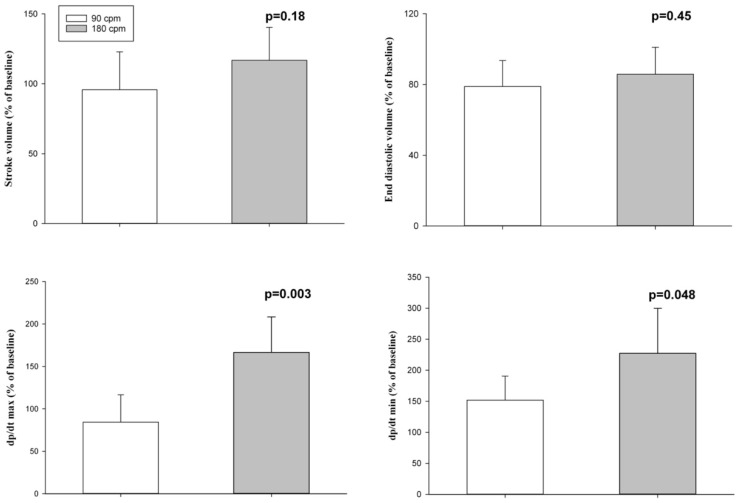
% change from baseline in stroke volume, end-diastolic volume, dp/dt max (maximal rate of rise of left ventricular pressure), and dp/dt min (minimum rate of change of ventricular pressure).

**Table 1 children-09-01838-t001:** Characteristics of Newborn Piglets at Baseline and at Commencement of Cardiopulmonary Resuscitation.

	CC Rate 90/min (n = 7)	CC Rate 180/min (n = 7)	*p* Value
Baseline characteristics			
Age (days)	2 (1–3)	3 (1–3)	0.81
Weight (kg)	2.2 (2.0–2.4)	2.0 (1.9–2.2)	0.21
Heart rate (bpm)	144 (142–159)	140 (131–168)	0.86
Mean Arterial blood pressure (mmHg)	53 (52–61)	57 (50–62)	0.82
Carotid flow (mL/min)	36 (34–51)	31 (20–37)	0.18
Cerebral oxygenation (%)	32 (32–41)	36 (34–45)	0.20
pH	7.47 (7.46–7.51)	7.51 (7.48–7.52)	0.12
PaO_2_ (torr)	62 (61–90)	62 (55–71)	0.46
PaO_2_ (torr)	36.3 (35.0–38.9)	37.4 (34.1–40.0)	0.94
Base excess (mmol/L)	3 (2–4)	4 (3–7)	0.17
Lactate (mmol/L)	3.6 (2.5–4.2)	2.6 (2.5–3.1)	0.37
Duration of asphyxia (s)	440 (280–506)	470 (360–585)	0.65
Characteristics at commencement of Resuscitation			
Heart rate (bpm)	0 (0–0)	0 (0–0)	
Carotid blood flow (mL/min)	0 (0–0)	0 (0–0)	
Arterial pH	6.58 (6.54–6.68)	6.55 (6.50–6.73)	0.46
paCO_2_ (torr)	102 (67–121)	106 (86–112)	0.68
Lactate (mmol/L)	16 (16–19)	19 (17–20)	0.13
Base Excess (mmol/L)	−29 (−30–−26)	−28 (−30–−22)	0.65
Characteristics immediately after return of spontaneous circulation			
Heart rate (bpm)	180 (165–212)	204 (182–248)	0.21
Carotid blood flow (mL/min)	23 (21–30)	20 (16–23)	0.12
Arterial pH	6.78 (6.52–6.84)	6.78 (6.74–6.96)	0.54
paCO_2_ (torr)	50 (39–73)	52 (40–58)	0.43
Lactate (mmol/L)	20 (18–20)	20 (20–20)	0.36
Base Excess (mmol/L)	−28 (−30–−26)	−26 (−29–−25)	0.38
Characteristics 30 min after return of spontaneous circulation			
Heart rate (bpm)	190 (152–219)	206 (192–241)	0.19
Carotid blood flow (mL/min)	29 (25–39)	33 (18–39)	0.87
Arterial pH	7.02 (6.93–7.19)	7.12 (6.97–7.24)	0.65
paCO_2_ (torr)	37 (33–57)	33 (28–44)	0.64
Lactate (mmol/L)	16 (14–19)	17 (16–19)	0.47
Base Excess (mmol/L)	−20 (−23–−15)	−20 (−23–−17)	0.79

Data are presented as median (IQR); CC = chest compression.

**Table 2 children-09-01838-t002:** Respiratory parameters during CPR.

	CC Rate 90/min (n = 7)	CC Rate 180/min (n = 7)	*p* Value
Tidal volume (mL/kg)	5.8 (0.9)	5.3 (1.4)	0.477
Minute Ventilation (mL/kg/min)	522 (79)	945 (248)	0.003
Peak Inspiratory Flow (L/min)	3.7 (0.4)	5.8 (0.9)	0.0007
Peak Expiration Flow (L/min)	−5.8 (0.7)	−8.0 (0.7)	0.0009
Peak Inflation Pressure (cmH_2_O)	30.0 (1.5)	31.1 (4.2)	0.546
Positive End Expiratory Pressure (cmH_2_O)	29.1 (2.0)	30.2 (5.1)	0.625
End-tidal CO_2_ (mmHg)	18 (7)	23 (15)	0.248
Rate (/min) *	90 (1)	179 (1)	<0.0001

Data are presented as the mean (SD), ***** Rate = Ventilation and number of chest compressions, which corresponds to the number of ventilations per minute; CC = chest compression.

**Table 3 children-09-01838-t003:** Data separated by sex.

	CC Rate 90/min (n = 7)			CC Rate 180/min (n = 7)		
	Male (n = 2)	Female (n = 5)	*p* Value	Male (n = 3)	Female (n = 4)	*p* Value
Asphyxia time (s) ^#^	399 (198–600)	440 (313–501)	1.00	430 (360–585)	481 (365–573)	0.86
Achieving ROSC (n)	1 (50%)	3 (60%)	1.00	3 (100%)	4 (100%)	1.00
ROSC time (s) ^#^	395 (189–600)	136 (88–600)	0.57	79 (73–118)	137 (95–290)	0.23
Requiring epinephrine (n)	2 (100%)	3 (60%)	1.00	0 (0%)	2 (50%)	0.43
Epinephrine doses ^#^	2 (1–3)	1 (0–3)	0.57	0 (0–0)	0.5 (0–2)	0.40

Data are presented as numbers (%), unless indicated ^#^ median (IQR); CC = chest compression.

## Data Availability

The authors declare that all supporting data are available within the article.

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
