# Peer review of "Chest Compression Rates of 90/min versus 180/min during Neonatal Cardiopulmonary Resuscitation: A Randomized Controlled Animal Trial"

_children, 2022, doi:10.3390/children9121838_

Round 1
Reviewer 1 Report
Bruckner and colleagues report on a small cohort of asphyxiated newborn piglets, in whom they compared a chest compression rate at 90/min and 180 /min, in order to evaluate the time to return to ROSC, survival, hemodynamics and respiratory parameters. They found that survival and time to ROSC were improved with the chest compression rate at 180/min but the difference was not statistically significant. Instead hemodynamic and respiratory parameters were improved and the difference was statistically significant. The study is well designed and well written. The manuscript contains an appropriate amount of technical information to describe and illustrate the methods used, and the data analysis and discussion are adequate.
In the current study, during the resuscitation of the asphyxiated newborn piglets, sustained lung inflation for 30 s was provided in both groups of piglets that received a chest compression rate of 90/min and 180/min respectively. This is currently not recommended by the neonatal resuscitation guidelines. Although the authors acknowledged this limitation, this is a bias of the study.
Authors should have explored the effect on survival and time to ROSC during chest compression at a rate of 90/min of sustained inflations compared to the standard positive pressure ventilation (one ventilation every three compression). Then, the two different chest compression rate during sustained inflation should have been compared, given the fact that the standard positive pressure ventilation (one ventilation every three compressions) is nearly impossible with a chest compression rate of 180/min.The authors should discuss this issue in more detail.
Author Response
Bruckner and colleagues report on a small cohort of asphyxiated newborn piglets, in whom they compared achest compression rate at 90/min and 180 /min, in order to evaluate the time to return to ROSC, survival, hemodynamics and respiratory parameters. They found that survival and time to ROSC were improved withthe chest compression rate at 180/min but the difference was not statistically significant. Instead hemodynamicand respiratory parameters were improved and the difference was statistically significant. The study is welldesigned and well written. The manuscript contains an appropriate amount of technical information to describeand illustrate the methods used, and the data analysis and discussion are adequate.
Response: Thank you
In the current study, during the resuscitation of the asphyxiated newborn piglets, sustained lung inflation for 30 s was provided in both groups of piglets that received a chest compression rate of 90/min and 180/minrespectively. This is currently not recommended by the neonatal resuscitation guidelines. Although the authorsacknowledged this limitation, this is a bias of the study.
Response: Thank you, we agree that this is not currently recommended in the neonatal resuscitationguidelines. We would like to also acknowledge that the authors Schmölzer and Cheung are the PIs ofthe SURV1vE-trial, which is comparing 3:1 C:V versus the sustained inflation + chest compressionapproach during neonatal resuscitation in the delivery room.
Authors should have explored the effect on survival and time to ROSC during chest compression at a rate of90/min of sustained inflations compared to the standard positive pressure ventilation (one ventilation everythree compression). Then, the two different chest compression rate during sustained inflation should havebeen compared, given the fact that the standard positive pressure ventilation (one ventilation every threecompressions) is nearly impossible with a chest compression rate of 180/min. The authors should discuss thisissue in more detail.
Response: Thank you, we have tried to discuss this in more detail in the paper. In our originalpublication (Schmölzer et al, Circulation 2013) we compared 3:1 C:V with CC+SI using a chestcompression rate of 120/min. We used a chest compression rate of 120/min, as the neonatalresuscitation guidelines recommend 120 action per minute (with 3:1 C:V = 90 chest compression and30 inflations).
This study showed that with CC+SI the i) time to ROSC was 38 (23–44)sec vs. 143 (84–303)sec(p=0.0008), ii) survival 7/8 [87.5%] vs. 3/8 [37.5%] (p=0.038), iii) 100% oxygen 3/8 vs. 8/8 (p=0.0042), andepinephrine use 0/8 vs. 7/8 (p<0.0001) compared to 3:1 C:V.
We have then compared CC+SI with either a chest compression rate of 90/min or 120/min (Li et al, PlosOne 2016) and observed similar time of ROSC, survival rates, hemodynamic and respiratoryparameters during cardiopulmonary resuscitation.
We then moved on to compared 3:1 C:V with CC+SI at a rate of 90/min (Li et al, Neonatology 2017). CC+SI significantly reduced the median (IQR) time of ROSC, i.e., 34 s (28-156 s) versus 210 s (72-300 s) in the 3:1 C:V group (p = 0.048). CC+SI also significantly reduced the requirement for 100% oxygen, improved respiratory parameters, and resulted in a similar hemodynamic recovery.
We have tried to integrate all these studies to hopefully improve the path of our study.
Reviewer 2 Report
The technical methods are quite good throughout this well designed study. However, the statistical analysis is not. While the main data were non-linear, the medians were analyzed as means. This introduces a strong impact from outliers from the normal distribution. Possibly more important, the authors do not describe how they have treated missing data. Three subjects in one treatment group failed to achieve ROSC. It is not clear whether these subjects were excluded from the comparison of time to ROSC, or their times to ROSC were replaced by 10 minutes, as suggested in the ranges reported Table 3. That would be statistically inappropriate. The authors must explain what they did in this respect. Furthermore, one important result is in terms of count data (survival). The authors do not report how they analyzed count data statistically.
Moreover, only a few minor outcomes were documented as statistically significant. Nonetheless the authors repeatedly state that more important outcomes (time to ROSC, survival) were improved. It is not sufficient to state in a subordinate clause that these results were not statistically significant. It is not defensible to state, e.g. in the Conclusion, "... time to ROSC and survival were improved ...". Statistically, they were not. That means they were not. The reported differences could be attributed to chance. It is acceptable to describe differences that nearly miss statistical significance (p > 0.05, p < 0.10) as "trends", but not as differences. The difference in time to ROSC might be a trend, though the method of calculation (mean of non-linear data) is inappropriate and treatment of missing data is unclear, so that cannot be ascertained here. The difference in survival rates, with a p of 0.46 by an unreported method, p of 0.19 by Fisher's exact test, is neither statistically significant nor a trend.
Statistical methods must be explained more clearly, perhaps revised, and in any case respected.
Some detailed suggestions and comments are included in the attachment.

Author Response
Reviewer #2:
The technical methods are quite good throughout this well designed study. However, the statistical analysis isnot. While the main data were non-linear, the medians were analyzed as means. This introduces a strongimpact from outliers from the normal distribution. Possibly more important, the authors do not describe howthey have treated missing data. Three subjects in one treatment group failed to achieve ROSC. It is not clearwhether these subjects were excluded from the comparison of time to ROSC, or their times to ROSC werereplaced by 10 minutes, as suggested in the ranges reported Table 3. That would be statistically inappropriate. The authors must explain what they did in this respect. Furthermore, one important result is in terms of countdata (survival). The authors do not report how they analyzed count data statistically.
Response:
Moreover, only a few minor outcomes were documented as statistically significant. Nonetheless the authorsrepeatedly state that more important outcomes (time to ROSC, survival) were improved. It is not sufficient tostate in a subordinate clause that these results were not statistically significant. It is not defensible to state, e.g. in the Conclusion, "... time to ROSC and survival were improved ...". Statistically, they were not. That meansthey were not. The reported differences could be attributed to chance. It is acceptable to describe differencesthat nearly miss statistical significance (p > 0.05, p < 0.10) as "trends", but not as differences. The difference intime to ROSC might be a trend, though the method of calculation (mean of non-linear data) is inappropriateand treatment of missing data is unclear, so that cannot be ascertained here. The difference in survival rates,with a p of 0.46 by an unreported method, p of 0.19 by Fisher's exact test, is neither statistically significant nora trend.
Response:
Statistical methods must be explained more clearly, perhaps revised, and in any case
respected.
Response: This has been added
Some detailed suggestions and comments are included in the attachment.
Response: These are answered below
Asynchronous ventilation (without pausing chest compressions) may be beneficial, but it does not conform to the protocol involving interposed ventilations the study undertakes to investigate. Effectively interposing ventilations may plausibly be affected by chest compression rate. This warrants discussion.
Response: we agree that asynchronous ventilation (without pausing chest compressions) may be beneficial and it is used in pediatric patients. We have studied our technique for over a decade now and the technique basically just uses a high distending pressure like CPAP (we called that sustained inflation more than 10 years ago, at a time when sustained inflation was an approach many investigators examined in preterm infants). With this technique a constant high distending pressure is provided and with each compression of the chest air is pushed out of the lung, and during each chest recoil air passively flows back into the lung. Indeed, the VT passively flowing into the lung = the air pushed out of the lung. While with either 3:1 or asynchronous ventilation (without pausing chest compressions) there is always more air lost during chest compression and delivered with each inflation.
You identify all of the other apparatus (ventilator, monitors, sensors, even catheters by brand name and model. But, not the chest compressor, which is central to the study. Either identify the commercial device that was used, or explain that a custom device was used and describe it.
Response: we have added custom made device in our lab.
cm/s2 should be revised, with the '2' as a superscript (what is meant is centimeters per squared seconds)
Response: This has been edited
Not all subjects achieved ROSC. You need a clause explaining how long you persisted in resuscitation efforts if ROSC was not achieved. This detail is at the end of the next sentence, in passive voice. It needs to be here, in active voice.
Response: this has been edited
This is the first time this is mentioned. It needs to be introduced in the Methods section.
Response: this has been edited
The authors are reporting data from an unidentified study - inappropriate.
Response: this study has been published since and we have added the reference.
What does this superscripted '29' mean. It is higher than any reference number.
Response: This was an error from the reference program, which has been edited.
You have not discussed how you treated missing data statistically. 3 of 7 subjects in the 90 CC/min group did not achieve ROSC and so could not contribute measured time to ROSC. By the reported ranges (upper limit 600 seconds), it appears that you have used 10 minutes for these subjects' time to ROSC. That would be arbitrary, and inaccurate. It would inflate the metric for the 90/min group, particularly when you take the mean, rather than the median, of the results.
Response: We have added this to the static section. We used the 10 minute max CPR time and sued this a time to ROSC. We disagree, if you would run a clinical trial and time to ROSC is your primary outcome, and an infant die, you would not exclude that one from the analyses and would add the duration of CPR to your analysis.
Data separated by sex are presented in Table 3. Reviewers question was why?
response: There are be male and female sex differences in cardiovascular research reported in adults. There is also a call to publish outcomes as supplements by sex and gender (were applicable). This is a standard requirement in all Canadian grant applications and we have a long standing history to do this with our data.
Why would you calculate the mean of the median? Justify this anomalous approach.
Response: this is not an anomalous approach, we have used this approach since 15 years or so. This was actually advised by one of our statisticians, this approach allows to only have 1 value per piglets for the duration of the resuscitation instead of using each inflation throughout the resuscitation. This approach will eliminate every obvious significant differences just by the sheer number of inflations being analyzed.
Reviewer 3 Report
Introduction:
- when the normal heart rate ranges from 110-160/min in term fetuses and human neonates, why are the authors investigating a supranormal heart rate of 180/min? Why are they not investigating 120 or even upto 160/min?
Methods:
- why did the authors choose to use sustained inflation in the control group instead of the standard of care of 90 compressions coordinated with 30 breaths? Kindly elaborate.
Experimental protocol:
Why did the authors wait only 15 seconds after establishment of asystole? Did any piglets recover with positive pressure ventilation alone?
Lines 193-198 are not clear. did the piglet receive manual chest compressions or automated chest compression?
Line 194: why was 21% oxygen used during chest compressions when the current recommendations are to use 100% oxygen?
Line 197: explain to the readers if that 1 second interruption in sustained inflation is adequate for exhalation.
Line 201: consider adding reference for 3-mL saline flush (https://pubmed.ncbi.nlm.nih.gov/34205843/ , https://pubmed.ncbi.nlm.nih.gov/33687959/)
Line 234: 91(34) is missing unit (sec)
Lines 270-271: this would be important to include in introduction section as explanation for choosing a high rate of 180/min in this study.
The obvious question arises that will a high rate of compressions of 180/min allow for sufficient time for filling of the heart during the decompression phase.
Line 285: Vali et al in https://www.ncbi.nlm.nih.gov/pmc/articles/PMC8286977/ showed higher carotid blood flow and improved cerebral oxygen delivery with continuous chest compressions at 120/min, which may add to the discussion in this manuscript.
Limitations:
- the authors mention about ET tube, however, the current study utilized tracheostomy.
- When sustained inflation is currently not recommended, why did the authors use SI instead of continuous chest compressions with asynchronous ventilation. This is a major limitation that needs to be mentioned first ahead of the other sentences in the limitation section.
- 15 seconds is a very short period of asystole.
- is use of automated chest compression machine even feasible in the delivery room setting for human neonates?
- Authors did not look at longer term outcomes beyond the short study period of 1 hour- such as MRI brain changes.
- Brain, lung and heart tissues were not evaluated.
- was cerebral oxygen delivery compared between the groups?
Figure 1 mentions only n=5 in 90/min CC group in contrast to n=7 mentioned in manuscript text.
Author Response
Reviewer #3:
Introduction:
- when the normal heart rate ranges from 110-160/min in term fetuses and human neonates, why are the authors investigating a supranormal heart rate of 180/min? Why are they not investigating 120 or even upto 160/min?
Response: Thanks, as per reviewer #2 suggestions we have complete rewritten the introduction to justify our study in more detail
Methods:
- why did the authors choose to use sustained inflation in the control group instead of the standard of care of 90 compressions coordinated with 30 breaths? Kindly elaborate.
Response: we have used this technique for the last 10 years and have done several animal studies as well as have completed now two randomized trials in newborn infants. One has been published several years ago and the SURV1VE-trial has just been completed and we are currently analyzing the data. This were the reasons why we have use this technique.
Experimental protocol:
Why did the authors wait only 15 seconds after establishment of asystole? Did any piglets recover with positive pressure ventilation alone?
Response: No piglet had ROSC with PPV alone. All these piglet s have 0 heart rate and 0 carotid blood flow, and we have done previous studies were we waited for 30 sec and even 60sec. In these circumstances, and the rate of ROSC is significantly less the longer you wait.
Lines 193-198 are not clear. did the piglet receive manual chest compressions or automated chest compression?
Response: The piglets received automated chest compression, to i) allow consistent rates and ii) reduce any bias. There have been several publications about our automated chest compression device by Bruckner et al (pubmed: 35135850, 33541920, 34330756)
Line 194: why was 21% oxygen used during chest compressions when the current recommendations are to use 100% oxygen?
Response: Thanks, while this is not recommended in the guidelines, we have used 21% oxygen since several years in our chest compression studies, based on a meta-analysis we conduced in 2018. A Review of Oxygen Use During Chest Compressions in Newborns-A Meta-Analysis of Animal Data. Garcia-Hidalgo C, Cheung PY, Solevåg AL, Vento M, O'Reilly M, Saugstad O, Schmölzer GM. Front Pediatr. 2018 Dec 18;6:400. doi: 10.3389/fped.2018.00400. eCollection 2018.
Line 197: explain to the readers if that 1 second interruption in sustained inflation is adequate for exhalation.
Response: The one second is enough for exhalation, however, with each chest compression you also exhale, which is the difference in our technique.
Line 201: consider adding reference for 3-mL saline flush (https://pubmed.ncbi.nlm.nih.gov/34205843/ , https://pubmed.ncbi.nlm.nih.gov/33687959/)
Response: This has been added
Line 234: 91(34) is missing unit (sec)
Response: This has been added
Lines 270-271: this would be important to include in introduction section as explanation for choosing a high rate of 180/min in this study.
Response: as per reviewer #2 suggestions we have complete rewritten the introduction to justify our study in more detail
The obvious question arises that will a high rate of compressions of 180/min allow for sufficient time for filling of the heart during the decompression phase.
Response: Thanks, while not significantly improved, stroke volume was improved with rate of 180/min. In addition, left ventricular functions were also improved with the higher rate.
Line 285: Vali et al in https://www.ncbi.nlm.nih.gov/pmc/articles/PMC8286977/ showed higher carotid blood flow and improved cerebral oxygen delivery with continuous chest compressions at 120/min, which may add to the discussion in this manuscript.
Thanks, this has been added.
Limitations:
- the authors mention about ET tube, however, the current study utilized tracheostomy.
Response: thanks, this has been added
- When sustained inflation is currently not recommended, why did the authors use SI instead of continuous chest compressions with asynchronous ventilation. This is a major limitation that needs to be mentioned first ahead of the other sentences in the limitation section.
Response: continuous chest compressions with asynchronous ventilation is also not recommend by NRP, however, we have rearranged the limitation section.
- 15 seconds is a very short period of asystole.
Response: this has been added
- is use of automated chest compression machine even feasible in the delivery room setting for human neonates?
Response: I believe it is feasible with the right technology and equipment.
- Authors did not look at longer term outcomes beyond the short study period of 1 hour- such as MRI brain changes.
Response: thanks, we did not
- Brain, lung and heart tissues were not evaluated.
Response: no we did not, in many of our previous experiments we did analyses heart, lung, and brain tissues and never found any differences, therefore we omitted it in this experiment.
- was cerebral oxygen delivery compared between the groups?
Response: We measured cerebral oxygenation, however we use an INVOS device, which basically only provides values as low as 15. During asystole and CPR, these values are always around 15 and therefore we decided not to include them as we thought there is no additional value in these numbers.
Figure 1 mentions only n=5 in 90/min CC group in contrast to n=7 mentioned in manuscript text.
Response: yes, only 5 piglets achieved ROSC from the 7 who were randomized to this group.
Round 2
Reviewer 2 Report
The authors have substantially and satisfactorily addressed my concerns except in one instance. The remaining problem is that the reported time to ROSC for the 90/min group continues to include an arbitrary 600 s for the 2 subjects that were not resuscitated. This artificially inflates the time to ROSC for that group. If the protocol had included sustaining efforts for 20 minutes as opposed to 10, the mean for that group would have nearly doubled (because of the outsized impact of outliers in a mean). Instead, in principal they should report the number achieving ROSC and the time to ROSC among those that achieved it. Using a median for averaging the results would improve robustness to this distortion.
However, by including other improvements in clarity (specifying that they included the non-resuscitated subjects and that they used specifically 10 minutes in the calculation of time to ROSC for the non-resuscitated subjects) they have made their treatment sufficiently clear to be acceptable, particularly since they no longer suggest a statistically significant difference. The manuscript could be improved by reporting the time to ROSC of only those that achieved ROSC, but it is now sufficiently clear to be acceptable.
Author Response
Thank you,
We have edits dated result section and now present:
1) median (IQR) for all piglets how achieved ROSC or did not achieve ROSC combined
2) Have removed the piglets two did not achieve ROSC and provided the median (IQR) for them as well.